# Thoracic outlet syndrome (TROTS) registry: A study protocol for the primary upper extremity deep venous thrombosis section

**Ludo Schropp**[1], **Robert J. C. M. F. de Kleijn**[1], **Jan Westerink**[2], **Mathilde Nijkeuter**[3], **Evert-Jan Vonken**[4], **Irene C. van der Schaaf**[4], **H. Stephan Goedee**[5], **Alexander F. J. E. Vrancken**[5], **Eline S. van Hattum**[1], **Bart-Jeroen Petri**[1], **Gert J. de Borst**[1]*, **TROTS registry collaborators**[¶]

1 Department of Vascular Surgery, University Medical Center Utrecht, Utrecht, The Netherlands,
2 Department of Internal Medicine, Isala Hospital, Zwolle, The Netherlands, 3 Department of Vascular Medicine, University Medical Center Utrecht, Utrecht, The Netherlands, 4 Department of Radiology, University Medical Center Utrecht, Utrecht, The Netherlands, 5 Department of Neurology and Neurosurgery, UMC Utrecht Brain Center University Medical Center Utrecht, Utrecht, The Netherlands

¶ The TROTS registry collaborators are listed in the acknowledgments.
* G.J.deborst-2@umcutrecht.nl

**Data Availability Statement:** No datasets were generated or analysed during the current study. All

# Abstract

## Introduction

There is a lack of comprehensive and uniform data on primary upper extremity deep venous thrombosis (pUEDVT). pUEDVT includes venous thoracic outlet syndrome related upper extremity deep venous thrombosis (UEDVT) and idiopathic UEDVT. Research on these conditions has been hampered by their rarity, lack of uniform diagnostic criteria, and heterogeneity in therapeutic strategies. To improve current research data collection using input of all various pUEDVT treating medical specialists, we initiated the ThoRacic OutLet Syndrome (TROTS) registry. The aim of the TROTS registry is to a) collect extensive data on all pUEDVT patients through a predefined protocol, b) give insight in the long term outcome using patient reported outcome measures, c) create guidance in the diagnostic and clinical management of these conditions, and thereby d) help provide content for future research.

## Methods and analysis

The TROTS registry was designed as an international prospective longitudinal observational registry for data collection on pUEDVT patients. All pUEDVT patients, regardless of treatment received, can be included in the registry after informed consent is obtained. All relevant data regarding the initial presentation, diagnostics, treatment, and follow-up will be collected prospectively in an electronic case report form. In addition, a survey containing general questions, a Health-related Quality of Life questionnaire (EQ-5D-5L), and Functional Disability questionnaire (Quick-DASH) will be sent periodically (at the time of inclusion, one and two years after inclusion, and every five years after inclusion) to the participant. The registry protocol was approved by the Medical Ethical Review Board and registered in the Netherlands Trial Register under Trial-ID NL9680. The data generated by

relevant data from this study will be made available upon study completion.

**Funding:** The authors received no specific funding for this work.

**Competing interests:** The authors have declared that no competing interests exist. TROTS registry collaborators: We have read the journal's policy and three of the TROTS registry collaborators have the following competing interests: Vincent Jongkind has received financial support for research from Medtronic. Michiel Coppens has received financial support for research from Bayer, Roche, UniQure, and Novo Nordisk; Honoraria for lecturing from Bayer, CSL Behring, and Alexion; Fees for consulting from Sobi, Viatris, CSL Behring, Novo Nordisk, and Daiichi Sankyo; Travel support for attending meeting: Alexion, Sobi. Marieke J.H.A. Kruip has received an unrestricted grant from Sobi, payment to institute. Speakers fee from Sobi, Roche, Bristol Myers Squibb, payment to institute. This does not alter our adherence to PLOS ONE policies on sharing data and materials.

the registry will be used for future research on pUEDVT and published in peer reviewed journals.

## Conclusion

TROTS registry data will be used to further establish the optimal management of pUEDVT and lay the foundation for future research and guidelines.

## Introduction

### Evidence gap

**Upper extremity deep venous thrombosis.**   Upper extremity deep venous thrombosis (UEDVT) can be divided in primary (pUEDVT) and secondary UEDVT (sUEDVT) [1]. sUEDVT, in which a clear cause for thrombosis can be identified (e.g. indwelling catheters, pacemaker wires, malignancy, etc.), is the most common variant accounting for almost 90% of all UEDVT cases [2, 3]. In contrast, in pUEDVT the cause for thrombosis is unknown (idiopathic UEDVT) or can occur as a result of venous thoracic outlet syndrome (VTOS). The management and pathophysiology of sUEDVT are well established and this condition is therefore not included in this registry. The management and pathophysiology of pUEDVT on the other hand, are not [4]. pUEDVT literature almost exclusively comprises retrospective, small, monocenter patient cohorts often treated and reported by one medical specialty. There are no prospective (randomized) series available comparing the various management regimens, resulting in a lack of universally accepted treatment recommendations and guidelines.

**Venous thoracic outlet syndrome.**   VTOS is characterized by repetitive positional compression of the subclavian vein (SV) in the thoracic outlet, primarily in the ventral costoclavicular space and sometimes in the retropectoral minor space. The ventral costoclavicular space is formed by the clavicle, first rib, and anterior scalene muscle. Compression of the SV occurs with certain positions of the arm, mostly hyper abduction, narrowing the aforementioned spaces. Due to this often repetitive compression intraluminal damage and scarring of the SV occurs, resulting in stenosis and eventually thrombosis [1, 5, 6]. VTOS is one of the three subtypes of thoracic outlet syndrome (TOS) along with neurogenic TOS (NTOS) and arterial TOS (ATOS). NTOS is considered to be the most common form of TOS (95% of TOS cases), although highly dependent on the applied NTOS definition, and ATOS the rarest of the three (1% of TOS cases) [7, 8].

VTOS causes pain, swelling, and discoloration of the arm either due to positional compression of the vein without thrombosis referred to as McCleery syndrome or thrombosis of the vein [8, 9]. Patients with VTOS related UEDVT may present with an acute thrombosis, acute-on-chronic thrombosis, or with chronic complaints. In some cases, a provoking factor in the form of heavy physical labor (sports, heavy lifting, etc) may have shortly preceded the thrombosis; this is often referred to as Paget-Schroetter syndrome or effort-thrombosis as subtype of VTOS related UEDVT [5]. In the acute-on-chronic cases, patients have had some form of complaints of their affected arm due to stenosis, but suddenly develop an acute worsening of symptoms based on a newly developed thrombosis. In VTOS patients with chronic symptoms, complaints are caused by a chronically thrombosed vein or severe post thrombotic damage (venous webs) often accompanied by a large collateral vein network. Interestingly, the degree of vascular damage does not always seem to correlate with patient symptoms in VTOS. Some

patients with a (chronically) thrombosed vein experience minor or no complaints, while some patients with a recanalized vein and positional compression have severe symptoms.

Diagnosing VTOS related UEDVT is based on two pillars: the intravascular thrombosis component and the extraluminal compression component [10]. The diagnosis of UEDVT is commonly made with duplex ultrasound. If duplex ultrasound is inconclusive, computed tomography venography (CTV) and magnetic resonance venography (MRV) can be considered. Venography is regarded as the gold standard for UEDVT diagnosis [11]. The diagnosis of the VTOS compression component is based on positional compression of the SV in the thoracic outlet. Commonly used imaging modalities are duplex ultrasound, CTV, MRV, and venography, performed with the arm in neutral and stress position (hyperabduction) to provoke the compression [10, 12, 13]. However, there are no widely accepted diagnostic criteria for the VTOS. In addition it is unclear to what extend venous compression is seen in healthy volunteers (some studies report in up to 60%) and what amount of compression should be considered pathological [14, 15].

Treatment is based on recanalization of the vein and thoracic outlet decompression (TOD) [7, 16–18]. Recanalization is achieved with anticoagulation and/or endovascular techniques: catheter directed thrombolysis (CDT), thrombosuction, and venoplasty in the acute phase and in severe chronic cases venoplasty and stenting. Stent placement should be considered as bail-out technique in the most severe cases and should only be performed after TOD to reduce the risk of stent failure [19]. TOD is commonly achieved through first rib resection and a release of the anterior scalene muscle to decompress the SV [20]. TOD can be performed through a transaxillary (Roos procedure), infraclavicular, supraclavicular, combined/paraclavicular, and in more recent years a (robot assisted) transthoracic approach [21–23]. For VTOS, a single supraclavicular incision is considered inadequate since it is often impossible to perform a complete ventral first rib resection and venolysis. Hence the supraclavicular is often combined with an infraclavicular incision for adequate decompression of the vein and allows for direct venous reconstruction in case of a occluded or a severely stenotic vein [5, 21]. The transaxillary approach is considered to have superior cosmetic results. There are no studies that compare the short and long term outcomes of the various surgical techniques. The most commonly applied treatment regimen for acute VTOS associated UEDVT is catheter directed thrombolysis (within 14 days of symptom onset) followed by staged TOD, ideally within 2 weeks of CDT, often combined with post-operative balloon venoplasty in case of a persisting stenosis, and treatment with anticoagulation (vitamin K antagonist or direct oral anticoagulants (DOAC)) for 3–6 months post thrombosis [16, 17, 24, 25].

**Idiopathic upper extremity deep venous thrombosis.**   Contrasting patients with VTOS related UEDVT, patients with idiopatic UEDVT (iUEDVT) have no obvious trigger, compression, or underlying disease causing the thrombosis. iUEDVT symptoms are comparable to VTOS related UEDVT symptoms: pain, swelling, and blue discoloration. The diagnosis of UEDVT can be established with duplex ultrasound, CTV, MRV, or venography [11]. iUEDVT is treated with anticoagulation therapy for at least 3 months, mainly vitamin K antagonists or DOAC [1, 26].

**Primary upper extremity deep venous thrombosis management.**   The incidence of pUEDVT is estimated to be around 2–3 cases per 100.000 people per year [2, 3, 27]. Although the true incidence should be considered unknown since the epidemiological data is scarce and either old, monocenter, or both. The most important long term complication of pUEDVT is the post thrombotic syndrome (PTS) [28]. Although a rare disease, patients are generally young and otherwise healthy individuals in whom PTS can have a major impact on their quality of life [6, 8, 29]. Therefore early and adequate management is essential. Indisputable evidence on the management of pUEDVT is still lacking and therefore international

guidelines remain ambiguous and, in some points, may even contradict each other [30–33]. Some clinicians argue that venous compression in the thoracic outlet is a common physiological finding and therefore do not consider compression as the (main) cause for pUEDVT. They therefore do not generally investigate whether compression is present. On the contrary, others advocate that venous compression, or VTOS, can be the (main) provocative factor for pUEDVT, and in the absence of a clear secondary cause, this should be investigated [4]. This is reflected by the complete lack of advice in the guidelines on in which patients compression of the SV should be investigated as potential provoking factor and subsequently how this compression should be investigated. This lack of a uniform management guideline, supported by all treating physicians, has led to a twin-track treatment approach, both aiming to balance between the occurrence of treatment complications and the incidence of PTS [34–36]. pUEDVT patients are either treated conservatively with DOAC, commonly without VTOS screening, or invasively through CDT and according to VTOS screening with or without TOD based on their treating physicians' specialism and preference. There are currently no prospective (randomized) cohorts comparing both treatment strategies [17, 34]. This might be an argument to not perform VTOS screening in the pUEDVT population, since the superiority of the invasive over the conservative treatment approach has not been established in comparative research. On the other hand, the reviews analyzing different pUEDVT treatment regimens suggest an improved functional outcome and a reduction of PTS occurrence in the invasively treated groups (PTS incidence ranging from 5–21%) compared to the conservative strategy (PTS incidence ranging from 25–67%) [4, 17, 18, 34]. The risk for thrombosis recurrence seems comparable between the two treatment regimens (6–11%), limited information is available regarding bleeding complications [34]. These findings do need to be interpreted with caution given the retrospective nature, heterogeneity, and poor methodological quality of the available studies. Given this absence of uniform diagnostic algorithms and treatment guidelines there is potentially a great deal of overlap between iUEDVT and VTOS related UEDVT in current practice. This might result in patients being labeled idiopathic and treated conservatively while there is in fact a (unexamined) VTOS compression causing the UEDVT or the other way around with patients receiving invasive treatment at the cost of potential complications while the UEDVT is in fact not caused by VTOS compression but be a physiological phenomenon instead [4].

## Rationale of the TROTS registry

In summary there is an urgent need for comprehensive, uniform data collection on pUEDVT with multidisciplinary specialist involvement to prevent over- and undertreatment of pUEDVT. Therefore, among other things, we initiated the ThoRacic OutleT Syndrome (TROTS) registry, a prospective multicenter international web-based registry. In addition to pUEDVT research, the TROTS registry was also designed for thoracic outlet syndrome research in general including neurogenic and arterial TOS and McCleery syndrome. This is however not the scope of this manuscript and will be outlined elsewhere.

Registry based research has been widely accepted for research on rare conditions and studies of heterogeneous patient populations, both the case in pUEDVT. In addition, registries use broad inclusion criteria and few exclusion criteria, this leads to data with great generalizability and gives the opportunity to study specific subgroups [37, 38].

The aim of the TROTS registry is to a) collect extensive data on all pUEDVT patients through a predefined protocol, b) give insight in the long term outcome using patient reported outcome measures (PROM's), c) create guidance in the diagnostic and clinical management of these conditions, and thereby d) help provide content for future (randomized) research.

**Table 1. Objectives of the TROTS registry with respect to primary upper extremity deep venous thrombosis.**

| Objective 1 | To provide data on the incidence. |
|---|---|
| Objective 2 | To identify patients at risk. |
| Objective 3 | To optimize diagnostic protocols. |
| Objective 4 | To provide insight in and optimize the different treatment strategies. |
| Objective 5 | To provide insight in the complications of the different treatment strategies. |
| Objective 6 | To provide insight in and optimize the outcome of the different treatment strategies using PROM's. |
| Objective 7 | To provide an extensive database which can be used to research specific and broad research questions. |
| Objective 8 | To provide data which can be used to design future studies. |
| Objective 9 | To facilitate recruitment of patients into future studies. |
| Objective 10 | To create a collaboration of all medical specialists involved. |

pUEDVT: idiopathic upper extremity deep venous thrombosis, PROM's: patient reported outcome measures.

The main objectives of the TROTS registry are summarized in Table 1 In addition to these objectives many different research questions may be investigated with the data generated by the registry. The TROTS registry is expressly intended as a collaboration between various medical specialties that treat pUEDVT. These specialties include but are not limited to vascular surgeons, vascular medicine specialists, internal medicine specialists, and (interventional) radiologists.

## Materials and methods

### Study design and objectives

The TROTS registry was designed as an international prospective longitudinal observational registry for extensive data collection on TOS and iUEDVT patients. Participation in the TROTS registry does explicitly not influence the local protocols and patient management. The TROTS registry is initiated by both the Department of Vascular Surgery and the Department of Internal Medicine of the University Medical Centre Utrecht (UMCU), a large tertiary referral university hospital in the Netherlands. The TROTS steering committee is presently formed by members of these departments. The physicians of participating centres can at all times consult the UMCU multidisciplinary expert panel (consisting of a vascular surgeon, a vascular medicine specialist, a (interventional)radiologist, and a neurologist) regarding a specific patient or refer a patient for analysis.

The registry is designed and implemented according to the principles of the International Council for Harmonisation (ICH) Good Clinical Practice (GCP) guideline and according to the European Good Data Protection Regulation (GDPR) [39, 40].

The Utrecht medical ethics committee has approved the study protocol (ID 21–326) and the registry has been registered in the Netherlands Trial Register under Trial ID NL9680 (https://www.trialregister.nl/trial/9680). All patients must give informed consent (IC) before inclusion in the registry. In line with the Dutch law and regulations ('Wet medisch-wetenschappelijk onderzoek met mensen'), the medical ethics committee waived the need for parental or guardian consent, since adolescents aged 16 years and older give their own consent. The aim of the registry is to include as many patients as possible, inclusion and follow-up will therefore be ongoing indefinite.

**Table 2. In- and exclusion criteria for the TROTS registry, primary upper extremity deep venous thrombosis specific.**

| Inclusion criteria |
| --- |
| • A positive diagnosis for any form pUEDVT, regardless of treatment. |
| • Aged 16 years or older. |
| • Subject and/or legal representative is/are able to read and comprehend the patient information folder and informed consent form. |
| • Patient and/or their legal representative signed informed consent. |
| Exclusion criteria |
| • Secondary Upper extremity deep venous thrombosis caused by a indwelling device or active malignancy. |
| • Subject and/or legal representative is unwilling or unable to sign complete or partial informed consent. |

pUEDVT: primary upper extremity deep venous thrombosis, TOS: thoracic outlet syndrome, iUEDVT: idiopathic upper extremity deep venous thrombosis

## Registry population, pUEDVT specific

All patients with either any form of VTOS related UEDVT or iUEDVT, regardless of treatment received, can be included in the registry. In addition, subjects must be 16 years or older, must be able to read and comprehend the patient information folder (PIF) and IC form, and must sign IC (Table 2).

## Recruitment and consent process

Individuals eligible for inclusion are identified by the treating physician in concurrence with the local principal investigator (PI). A PIF and IC form (S2 File) will be provided to the participant as well as verbal information by the local PI at the first hospital visit. If the individual wishes to participate, the IC form is dated and signed by the individual and local PI during the subsequent hospital visit. Upon inclusion in the TROTS registry, the participant is assigned a unique study ID generated by Castor EDC. Subjects can leave the registry at any time without stating any reason and without any consequences. Subjects can either chose to exclude their data from further research or to allow the data that was collected before withdrawal to be used in further research.

## Data collection and monitoring

All relevant clinical information regarding the patients, diagnosis, treatment, and follow-up will be manually extracted from the patients electronic health record and will be entered in a specially designed electronic case report form (eCRF) in Castor EDC by the local PI. (Fig 1) Castor EDC is a browser-based, metadata-driven EDC software solution and workflow methodology for building and managing online databases conform GCP standards [41]. The eCRF was designed using the reporting standards of the Society for Vascular Surgery for TOS [8]. Clinical data will be collected at baseline (time of inclusion) and during follow-up. If a patient is treated invasively (endovascular or surgical), the intervention form will be completed 30 days after the last intervention to score any potential complications.

In addition, as a PROM, a survey containing general questions, a Health-related Quality of Life questionnaire (EQ-5D-5L), and Functional Disability questionnaire (Quick-DASH) will be sent via Castor by email to the participant [42, 43]. This survey will be sent automatically at set points in time, namely: at the time of inclusion, one and two years after inclusion, and every five years after inclusion.

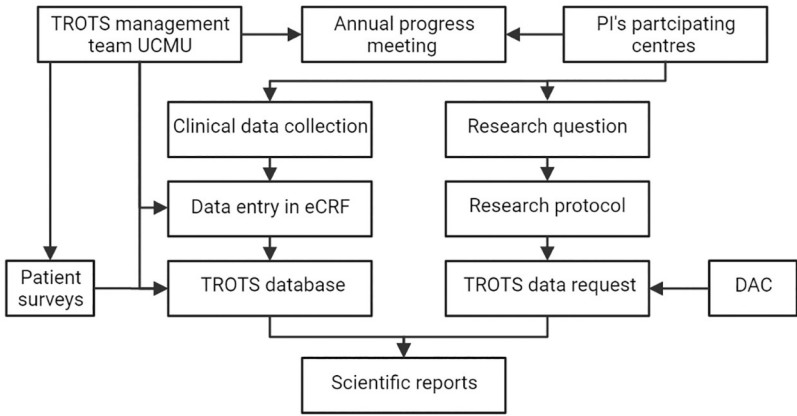

**Fig 1. Flowchart of TROTS registry structure.** TROTS: thoracic outlet syndrome, UMCU: University Medical Centre Utrecht, PI: principal investigator, eCRF: electronic case report form, DAC: data access committee.

An overview of the collected parameters is presented in Table 3 and the complete eCRF in the S4 File. Depending on new insights on pUEDVT these endpoints may be altered and updated, in such case this will be addressed in a research protocol amendment and presented to the medical ethical committee.

Within each participating centre, the local PI is responsible for the adequateness, accurateness, and completeness of the data entered in the eCRF. The data provided in the eCRF will be checked by the UMCU registry management team. (Fig 1) In case of inconsistencies, the local PI will be contacted and requested to adjust these matters. Annually, progress will be evaluated at a progress meeting between the steering committee and all local PI's of the participating centres.

## Patient and public involvement

The TROTS registry steering committee is currently exploring the possibilities with a major cardiovascular patient association in the Netherlands (Harteraad) to establish a pUEDVT patient association/network to improve patient involvement. There was currently no specific patient involvement in the design of the registry. The progress and results of the TROTS registry will be made publically available on the TROTS registry website www.TROTSregistry.com.

## Status and timeline of the TROTS registry

The TROTS registry was approved by the medical ethical committee June 2021 and directly started inclusion in the UMCU. In one year 76 patients with pUEDVT were included. At the time of writing of this manuscript, the registry is being expanded in multiple hospitals in the Netherlands (collaborators list, S1 File), aiming to connect with as many hospitals as possible. After implementation and finetuning of the registry in the Netherlands the next step will be expansion of the registry outside of the Netherlands.

## Ethical and safety considerations

As stated before, the TROTS registry is designed and implemented according to the ICH GCP guideline and according to the GDPR. After obtaining IC the IC form is stored at a secure location at the participating center and a unique study ID is generated. The key file containing the

**Table 3. Overview of the main collected parameters in the TROTS registry.**

| |
|---|
| The main parameters of interest include but are not limited to: |
| Patient demographics, characteristics, family history, and medical history, including: |
| • Age, length, and weight. |
| • Risk factors such as smoking or excessive upper extremity use (with sports or work) |
| • Medical history including prior trauma to the affected arm/thoracic outlet. |
| • Family history including thromboembolic disease, hypercoagulable state, and TOS. |
| Initial clinical assessment including the anamnesis and physical exam, including: |
| • Date of onset of symptoms |
| • Contralateral symptoms of the upper extremity |
| • Specific TOS tests (e.g. Roos test, upper limb tension test, etc) |
| Diagnostic modalities used and their results (including contralateral assessment if applicable), including: |
| • Laboratory tests including D-dimer and hypercoagulability testing |
| • (Dynamic) Plethysmography |
| • Electromyography |
| • Local muscle injection with anaesthetics or Botox |
| • (Dynamic) (duplex) ultrasound, CT, and MRI scans |
| • (Dynamic) Angiography and venography |
| Treatment(s) performed and treatment characteristics, including: |
| • Conservative measurements such as compression stockings and physical therapy |
| • Antithrombotic therapy, initial, maintenance, and periprocedural |
| • Local muscle injection with Botox |
| • Surgical treatment |
| • Endovascular interventions |
| Any treatment related complications, including: |
| • Complications related to antithrombotic treatment |
| • Short (<30 days) and long term (>30 days) complications surgery related |
| • Short (<30 days) and long term (>30 days) complications related to endovascular treatment |
| Follow-up data, if applicable, including: |
| • Symptoms during follow-up |
| • Physical exam during follow-up |
| • Diagnostics performed during follow-up and detailed description of results |
| • Change of treatment strategy, including discontinuation of medical treatment, and a detailed description of these treatments |
| • Changes in health-status such as diagnosis of a hypercoagulable state, malignancy, or any other diseases |
| • New contralateral diagnosis of TOS or iUEDVT |
| Follow-up survey sent at set time points during follow-up. Including: |
| • General health related questions |
| • Health-related Quality of Life questionnaire (EQ-5D-5L) |
| • Functional Disability of the arm questionnaire (Quick-DASH) |

TOS: thoracic outlet syndrome, CT: computed tomography, MRI: magnetic resonance imaging, iUEDVT: idiopathic upper extremity deep venous thrombosis

participants personal information and study ID is stored at a secure location. Information entered in the eCRF in Castor will be encrypted using the unique study ID. Confidentiality will be maintained at all times, participant information will not be disclosed to third parties. Persons with access to the keyfile are the local PI, data manager, and if needed the government healthcare inspection.

According to the European GDPR, each participant included in a European hospital is specifically asked for IC to share their encrypted data with participating centers outside the European Union (EU) in case of a data request. Participants may choose to decline this which will be documented in the eCRF (S4 File). Such data requests by participating centers outside the EU will only be honored if comparable legislation is in place in the respective country.

In case the registry data is used for future research (this process is explained further under 'dissemination plan'), the data will be extracted from Castor and saved in a secure research folder. The research team that submitted a research proposal will be provided with the anonymized data through secure transfer. The research team is responsible for secure storage and adequate handling of the research data.

## Dissemination plan

Each participating center may submit a research proposal with TROTS registry data to the TROTS Data Access Committee (DAC) using a data request form. Before submission of the research proposal, the research team must obtain a favorable opinion of its local medical ethical committee. The DAC consists of three UMCU based members (vascular surgeon, internal medicine specialists, and interventional radiologist) and four, rotating each year, PI's from participating centers. The DAC shall review research proposals and shall determine whether it is: relevant, feasible with available data, not already previously been or currently being researched, and set up in accordance with the GCP guideline and GDPR (or similar legislation if outside EU). (Fig 1) The decision to honor a proposal shall be made unanimously by the DAC. The provided data may only be used to address the research question as stated in the approved research proposal. To be able to reproduce the research findings and to help future users understand and reuse the data all changes made to the raw data and all steps taken in the analysis will be documented in syntaxes and by using new versions of the database by the research team. The research data, including raw data and syntaxes, will be archived for 15 years by the research team upon publication of the data. The research team is responsible for the dissemination of the results by submission of the research results in a peer reviewed journal. (Fig 1) Upon publication of a paper, the used data from the TROTS registry will be made available without restrictions in a publically accessible repository. This must be included in the research proposal for the DAC to approve a data request. If a paper is published, a reference to this registry protocol will be made. The research team may divide the authorship of the paper as they see fit, each participating center will be offered one place per participating medical specialty on the collaborators list. If a collaborators list is not allowed by the journal, the TROTS working group will be mentioned in the acknowledgements. It shall be encouraged to present the obtained results with the TROTS registry data on international congresses.

## Discussion and conclusion

There is a paucity of high quality data in the field of pUEDVT research resulting in diverse recommendations in the guidelines that sometimes even conflict between guidelines. pUEDVT research is challenging due to its rarity and the involvement of multiple medical specialties. Historically, studies report retrospective, monocenter, and monodisciplinary data. The TROTS registry will create a multicenter, multidisciplinary research team to prospectively collect uniform data of pUEDVT patients. Multiple research questions can be investigated with the TROTS registry data to further establish the optimal management of pUEDVT and lay the foundation for future research and guidelines.

## Supporting information

**S1 File. The TROTS registry collaborators.**
(PDF)

**S2 File. The patient information form and informed consent form of the Thoracic Outlet Syndrome (TROTS) registry.**
(PDF)

**S3 File. The TROTS registry protocol version 1.0 as approved by the medical ethical review board Utrecht, The Netherlands.**
(PDF)

**S4 File. The TROTS registry electronic case report form.**
(PDF)

## Acknowledgments

The TROTS registry collaborators

**Lead author: Çağdaş Ünlü** MD PhD, Department of Vascular Surgery, Noordwest-Ziekenhuisgroep, Alkmaar, Netherlands. Cagdas.Unlu@nwz.nl

**Remy H.H. Bemelmans** MD PhD, Department of Internal Medicine, Ziekenhuis Gelderse Vallei, Ede, the Netherlands.

**Peter E. Westerweel** MD PhD, Department of Internal Medicine, Albert Schweitzer Hospital, Dordrecht, the Netherlands.

**Maarten Lijkwan** MD PhD, Department of Surgery, Albert Schweitzer Hospital, Dordrecht, the Netherlands.

**Anne C. Esselink** MD, Department of Internal Medicine, Canisius Wilhelmina Ziekenhuis, Nijmegen, The Netherlands.

**Aron S. Bode** MD PhD, Department of Surgery, Canisius Wilhelmina Ziekenhuis, Nijmegen, The Netherlands.

**Hinke Nagtegaal** Msc, Department of internal medicine, Noordwest-Ziekenhuisgroep, Alkmaar, The Netherlands.

**Arina ten Cate** MD PhD, Thrombosis Expertise Center, Maastricht University Medical Center and Cardiovascular Research Institute Maastricht (CARIM), Maastricht, The Netherlands.

**Jorinde van Laanen** MD, Department of Vascular Surgery, Maastricht University Medical Center, Maastricht, The Netherlands.

**Arian van der Veer** MD PhD, Department of pediatric hematology, Maastricht University Medical Center+, Maastricht, The Netherlands.

**Vincent van Weel** MD PhD, Department of Surgery, Meander Medical Center, Amersfoort, The Netherlands.

**Gerben C. Mol** MD, Department of Internal Medicine, Meander Medical Center, Amersfoort, The Netherlands.

**Jasper Florie** MD PhD, Department of Radiology, Meander Medical Center, Amersfoort, The Netherlands.

**Daniel R. Faber** MD PhD, Department of Internal Medicine, BovenIJ hospital, Amsterdam, the Netherlands.

**Jeroen K. de Vries** MD, Department of Internal Medicine, Antonius hospital, Sneek, The Netherlands.

**Robertus H.W. van de Mortel** MD, Department of Vascular Surgery, Antonius hospital, Nieuwegein, The Netherlands.

**Marijke Molegraaf** MD PhD, Department of Vascular Surgery, Isala Hospital, Zwolle, the Netherlands.

**Vincent Jongkind** MD PhD, Department of Surgery, Amsterdam UMC vrije Universiteit Amsterdam, Amsterdam, The Netherlands. Amsterdam Cardiovascular Sciences, Microcirculation, Amsterdam, The Netherlands. Department of Physiology, Amsterdam Cardiovascular Sciences, Vrije Universiteit Amsterdam, Amsterdam, The Netherlands.

**Kakkhee Yeung** MD PhD, Department of Surgery, Amsterdam UMC vrije Universiteit Amsterdam, Amsterdam, The Netherlands. Amsterdam Cardiovascular Sciences, Microcirculation, Amsterdam, The Netherlands. Department of Physiology, Amsterdam Cardiovascular Sciences, Vrije Universiteit Amsterdam, Amsterdam, The Netherlands.

**Michiel Coppens** MD PhD, Department of Vascular Medicine, Amsterdam Cardiovascular Sciences, Amsterdam University Medical Centers, University of Amsterdam, Amsterdam, The Netherlands.

**Koen E.A. van der Bogt** MD PhD, Department of vascular surgery, Haaglanden Medical Center, Den Haag, the Netherlands.

**Edith D. Beishuizen** MD PhD, Department of internal medicine, Haaglanden Medical Center, Den Haag, the Netherlands.

**C. Heleen van Ommen** MD PhD, Department of Pediatric Hematology, Sophia Children's Hospital ErasmusMC, Rotterdam, The Netherlands.

**Marieke J.H.A. Kruip** MD PhD, Erasmus MC department of hematology, Erasmus University Medical Center, Rotterdam, the Netherlands.

**Marie Josee E. van Rijn** MD PhD, Department of Vascular and Endovascular Surgery, Erasmus University Medical Center, Rotterdam, The Netherlands.

**Thomas van Bemmel** MD PhD, Department of internal medicine, Gelre hospitaal, Apeldoorn, The Netherlands.

**Peter L. Klemm** MD PhD, Department of Vascular Surgery, Gelre Hospitaal, Apeldoorn, The Netherlands.

**Marcel M.C. Hovens** MD PhD, Department of internal medicine, Rijnstate hospital, Arnhem, The Netherlands.

**Paul M. van Schaik** MD PhD, Department of Surgery, Rijnstate hospital, Arnhem, The Netherlands.

**Matthijs Eefting** MD PhD, Department of internal medicine, Ikazia Hospital, Rotterdam, the Netherlands.

**Anne M.E. van Well** MD, Department of vascular surgery, Ikazia Hospital, Rotterdam, the Netherlands.

**Roos C. van Nieuwenhuizen** MD, Department of Vascular Surgery, Onze Lieve Vrouwen Gasthuis, Amsterdam, the Netherlands.

**Sanne van Wissen** MD PhD, Department of Internal Medicine, Onze Lieve Vrouwen Gasthuis, Amsterdam, the Netherlands.

**Martine C.M. Willems** MD PhD, Department of Surgery, Flevoziekenhuis, Almere, The Netherlands.

**Judith P. Post** MD, Department of Internal Medicine, Flevoziekenhuis, Almere, The Netherlands.

**Fleur S. Kleijwegt** MD PhD, Department of Internal Medicine, Rode Kruis Hospital, Beverwijk, The Netherlands.

**Monique H. Suijker** MD, Department of Paediatric Haematology, University Medical Center Utrecht, Utrecht University, Utrecht, the Netherlands. Van Creveldkliniek, University Medical Center Utrecht, Utrecht, the Netherlands.

## Author Contributions

**Conceptualization:** Ludo Schropp, Robert J. C. M. F. de Kleijn, Jan Westerink, Mathilde Nijkeuter, Evert-Jan Vonken, Irene C. van der Schaaf, H. Stephan Goedee, Alexander F. J. E. Vrancken, Eline S. van Hattum, Bart-Jeroen Petri, Gert J. de Borst.

**Data curation:** Ludo Schropp.

**Investigation:** Ludo Schropp.

**Methodology:** Ludo Schropp, Robert J. C. M. F. de Kleijn, Jan Westerink, Mathilde Nijkeuter, Evert-Jan Vonken, Irene C. van der Schaaf, H. Stephan Goedee, Alexander F. J. E. Vrancken, Eline S. van Hattum, Bart-Jeroen Petri, Gert J. de Borst.

**Project administration:** Ludo Schropp.

**Software:** Ludo Schropp.

**Supervision:** Gert J. de Borst.

**Writing – original draft:** Ludo Schropp.

**Writing – review & editing:** Ludo Schropp, Robert J. C. M. F. de Kleijn, Jan Westerink, Mathilde Nijkeuter, Evert-Jan Vonken, Irene C. van der Schaaf, H. Stephan Goedee, Alexander F. J. E. Vrancken, Eline S. van Hattum, Bart-Jeroen Petri, Gert J. de Borst.

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
