## [Decision Letter · Decision Letter 0]

4 Oct 2022

PONE-D-22-21260Thoracic outlet syndrome (TROTS) registry: a study protocol for the primary upper extremity deep venous thrombosis sectionPLOS ONE

Dear Dr. Schropp,

Thank you for submitting your manuscript to PLOS ONE. After careful consideration, we feel that it has merit but does not fully meet PLOS ONE’s publication criteria as it currently stands. Therefore, we invite you to submit a revised version of the manuscript that addresses the points raised during the review process.

We look forward to receiving your revised manuscript.

Kind regards,

Leila Harhaus

Academic Editor

PLOS ONE

"The authors have declared that no competing interests exist.

TROTS registry collaborators: We have read the journal’s policy and three of the TROTS registry collaborators have the following competing interests: Vincent Jongkind has received financial support for research from Medtronic. Michiel Coppens has received financial support for research from Bayer, Roche, UniQure, and Novo Nordisk; Honoraria for lecturing from Bayer, CSL Behring, and Alexion; Fees for consulting from Sobi, Viatris, CSL Behring, Novo Nordisk, and Daiichi Sankyo; Travel support for attending meeting: Alexion, Sobi. Marieke J.H.A. Kruip has received an unrestricted grant from Sobi, payment to institute. Speakers fee from Sobi, Roche, Bristol Myers Squibb, payment to institute."

3. One of the noted authors is a group or consortium the TROTS registry collaborators. In addition to naming the author group, please list the individual authors and affiliations within this group in the acknowledgments section of your manuscript. Please also indicate clearly a lead author for this group along with a contact email address.

Reviewers' comments:

Reviewer's Responses to Questions

**Comments to the Author**

1. Does the manuscript provide a valid rationale for the proposed study, with clearly identified and justified research questions?

Reviewer #1: Yes

2. Is the protocol technically sound and planned in a manner that will lead to a meaningful outcome and allow testing the stated hypotheses?

Reviewer #1: Partly

3. Is the methodology feasible and described in sufficient detail to allow the work to be replicable?

Reviewer #1: Yes

4. Have the authors described where all data underlying the findings will be made available when the study is complete?

Reviewer #1: No

5. Is the manuscript presented in an intelligible fashion and written in standard English?

Reviewer #1: Yes

6. Review Comments to the Author

You may also provide optional suggestions and comments to authors that they might find helpful in planning their study.

Reviewer #1: The authors present a study protocol of a multicenter registry for venous thoracic outlet syndrome. The study is necessary as vTOS is a rare disease and multicenter data on epidemiology, diagnosis and treatment is lacking. The presentation of the study protocol is well written and structured. However, there are some questions that need to be addressed:

- provide a CRF (or if it’s a web based registry with electronic CRF at least the structure of variables and data points that will be assessed), primary / secondary endpoints? Table 3 is rather prosaic

- refer to classification / reporting standards more precisely

- differentiate between Paget von Schroetter and McCleary

- provide more information on epidemiology – coming from the TOS point of view: it’s a rare subform of TOS

- provide the state of evidence regarding treatment decisions (surgical/conservative) more clearly

- provide the state of evidence regarding type of intervention (endovascular, type of surgery) more clearly

- please show within the abstract and manuscript were data is made available to the public (it seems to be journal policy)

- provide the TROTS registry website address (I didn’t find it with google)

7. PLOS authors have the option to publish the peer review history of their article (what does this mean?). If published, this will include your full peer review and any attached files.

Reviewer #1: **Yes: **Nora F. Dengler

---

## [Author Response · Author response to Decision Letter 0]

2 Nov 2022

Response to Academic Editor

Dear dr. Harhaus, 

Thank you for appraisal of our manuscript and the opportunity to submit a revised version. 

Please find below our responses to your and the reviewer’s comments. 

1 - Please ensure that your manuscript meets PLOS ONE's style requirements, including those for file naming. 

We have adjusted the manuscript and file names accordingly. 

2 – New competing Interests statement: 

The authors have declared that no competing interests exist.

TROTS registry collaborators: We have read the journal’s policy and three of the TROTS registry collaborators have the following competing interests: Vincent Jongkind has received financial support for research from Medtronic. Michiel Coppens has received financial support for research from Bayer, Roche, UniQure, and Novo Nordisk; Honoraria for lecturing from Bayer, CSL Behring, and Alexion; Fees for consulting from Sobi, Viatris, CSL Behring, Novo Nordisk, and Daiichi Sankyo; Travel support for attending meeting: Alexion, Sobi. Marieke J.H.A. Kruip has received an unrestricted grant from Sobi, payment to institute. Speakers fee from Sobi, Roche, Bristol Myers Squibb, payment to institute. This does not alter our adherence to PLOS ONE policies on sharing data and materials.

3 - One of the noted authors is a group or consortium the TROTS registry collaborators. In addition to naming the author group, please list the individual authors and affiliations within this group in the acknowledgments section of your manuscript. Please also indicate clearly a lead author for this group along with a contact email address.

We have revised the manuscript as requested. 

Response to reviewer

Dear dr. Dengler, 

Many thanks for your thorough review of our manuscript, your comments were very valuable in improving the paper.

Please find below our point by point responses to your comments. 

Kind regards, 

The research team 

1 - provide a CRF (or if it’s a web based registry with electronic CRF at least the structure of variables and data points that will be assessed), primary / secondary endpoints? Table 3 is rather prosaic

We have added the complete eCRF as supplemental file (S4 File). 

2 - refer to classification / reporting standards more precisely

Thank you for this suggestion, we have added a reference to the reporting standards (page 5 lines 107 and 110). We do wish to emphasize that the reporting standards were written to harmonize TOS research and not so much as for clinical guidance (yet). We have implemented the reporting standards in our registry accordingly in the design of our eCRF (page 13 line 305). 

3 - differentiate between Paget von Schroetter and McCleary

We have differentiated between these two entities more clearly in the text. (page 5 line 109) We have also highlighted that McCleery syndrome is not part of this manuscript, since the focus is on upper extremity venous thrombosis, but is included in the registry. (page 9 line 212)

4 - provide more information on epidemiology – coming from the TOS point of view: it’s a rare subform of TOS

More epidemiological background is provided in the text from a TOS (page 5 lines 104-107) and thrombosis point of view (page 7 lines 166-168). 

5- provide the state of evidence regarding treatment decisions (surgical/conservative) more clearly

Thank you for this suggestion. This is the main challenge in the care for a primary upper extremity deep venous thrombosis: there is no methodologically sound evidence that provides guidance for surgical versus conservative treatment. As a result there are yet no guidelines with clear universal recommendations on the optimal treatment algorithm. We have highlighted the lack of level 1 guidance as a shortcoming (page 4 lines 91-95) and elaborated on the possible treatment decisions (pages 8 and 9 lines 190-198). 

6- provide the state of evidence regarding type of intervention (endovascular, type of surgery) more clearly

Thank you. Again, the technical potential has significantly improved over recent years but there is yet no high quality data on whether we indeed should apply all these intervention options in a standard manner. See reply to comment 5: there is level 1 guidance on type of intervention. We have elaborated on the type of interventions in VTOS, both surgical and endovascular. (pages 6 and 7 lines 141-155) 

7- please show within the abstract and manuscript were data is made available to the public (it seems to be journal policy)

We have added a section on this matter (page 18 lines 396-398). For registry data requests, new research proposals need to be written and approved by medical ethical review boards. In these research proposals, the data sharing policy needs to be satisfactory for the Data Access Committee to approve a data request. 

8- provide the TROTS registry website address (I didn’t find it with google)

The website is currently still under construction. The address will be www.TROTSregistry.com, we have added this to the manuscript (page 16 line 348)

---

## [Decision Letter · Decision Letter 1]

13 Dec 2022

Thoracic outlet syndrome (TROTS) registry: a study protocol for the primary upper extremity deep venous thrombosis section

PONE-D-22-21260R1

Dear Dr. Schropp,

We’re pleased to inform you that your manuscript has been judged scientifically suitable for publication and will be formally accepted for publication once it meets all outstanding technical requirements.

Kind regards,

Samuele Ceruti

Academic Editor

PLOS ONE

Additional Editor Comments (optional):

Reviewers' comments:

Reviewer's Responses to Questions

**Comments to the Author**

1. Does the manuscript provide a valid rationale for the proposed study, with clearly identified and justified research questions?

Reviewer #1: Yes

2. Is the protocol technically sound and planned in a manner that will lead to a meaningful outcome and allow testing the stated hypotheses?

Reviewer #1: Yes

3. Is the methodology feasible and described in sufficient detail to allow the work to be replicable?

Reviewer #1: Yes

4. Have the authors described where all data underlying the findings will be made available when the study is complete?

Reviewer #1: Yes

5. Is the manuscript presented in an intelligible fashion and written in standard English?

Reviewer #1: Yes

6. Review Comments to the Author

You may also provide optional suggestions and comments to authors that they might find helpful in planning their study.

Reviewer #1: The authors answered to my concerns in a satisfactory way. I recommend to have the website working at the time point of article publication.

7. PLOS authors have the option to publish the peer review history of their article (what does this mean?). If published, this will include your full peer review and any attached files.

Reviewer #1: **Yes: **Nora F. Dengler

---

## [Editor Report · Acceptance letter]

28 Dec 2022

PONE-D-22-21260R1 

Thoracic outlet syndrome (TROTS) registry: a study protocol for the primary upper extremity deep venous thrombosis section 

Dear Dr. Schropp:

I'm pleased to inform you that your manuscript has been deemed suitable for publication in PLOS ONE. Congratulations! Your manuscript is now with our production department. 

Kind regards, 

on behalf of

Dr. Samuele Ceruti 

Academic Editor

PLOS ONE